# The Baikal subtype of tick-borne encephalitis virus is evident of recombination between Siberian and Far-Eastern subtypes

Grigorii A. Sukhorukov[1]*, Alexey I. Paramonov[2], Oksana V. Lisak[2], Irina V. Kozlova[2], Georgii A. Bazykin[1,3]*, Alexey D. Neverov[4,5]*, Lyudmila S. Karan[5]

**1** Center of Life Sciences, Skolkovo Institute of Science and Technology, Skolkovo, Russia, **2** Laboratory of molecular Epidemiology and genetic diagnosis, Scientific Centre for Family Health and Human Reproduction Problems, Irkutsk, Russia, **3** Laboratory of Molecular Evolution, Kharkevich Institute for Information Transmission Problems of the RAS, Moscow, Russia, **4** HSE University, Moscow, Russia, **5** Department of Molecular Diagnostics, Central Research Institute for Epidemiology, Moscow, Russia

* grsukhorukov@gmail.com (GAS); g.bazykin@skoltech.ru (GAB); neva_2000@mail.ru (ADN)

## Abstract

Tick-borne encephalitis virus (TBEV) is a flavivirus which causes an acute or sometimes chronic infection that frequently has severe neurological consequences, and is a major public health threat in Eurasia. TBEV is genetically classified into three distinct subtypes; however, at least one group of isolates, the Baikal subtype, also referred to as "886-84-like", challenges this classification. Baikal TBEV is a persistent group which has been repeatedly isolated from ticks and small mammals in the Buryat Republic, Irkutsk and Trans-Baikal regions of Russia for several decades. One case of meningoencephalitis with a lethal outcome caused by this subtype has been described in Mongolia in 2010. While recombination is frequent in *Flaviviridae*, its role in the evolution of TBEV has not been established. Here, we isolate and sequence four novel Baikal TBEV samples obtained in Eastern Siberia. Using a set of methods for inference of recombination events, including a newly developed phylogenetic method allowing for formal statistical testing for such events in the past, we find robust support for a difference in phylogenetic histories between genomic regions, indicating recombination at origin of the Baikal TBEV. This finding extends our understanding of the role of recombination in the evolution of this human pathogen.

## Author summary

Tick-borne encephalitis is a serious and frequently deadly infectious disease. It is caused by a virus of the same name with genome composed of single-stranded RNA. The known genomes of this virus fall into three large regional groups: Europe, Siberia, or the Russian Far East. These groups have originated from a common ancestor several hundred or thousand years ago and were assumed to have evolved independently since then. This study shows that a previously described group of viruses obtained in the vicinity of Lake Baikal in Russia have a mosaic genome: some parts of it are more closely related to those of the Siberian group, while others, to the Far Eastern group. Such a pattern probably arose

**Data Availability Statement:** Python scripts used for calculation of GSAUC and alignments are available at https://github.com/gregoruar/tbev_rec.

All other relevant data are within the manuscript and its Supporting Information files.

**Funding:** This work was funded by the Russian Science Foundation (project no. 21-74-20160 to G. A.B.). A.D.N. was partially supported by the HSE University Basic Research Program. The funders had no role in study design, data collection and analysis, decision to publish, or preparation of the manuscript.

**Competing interests:** The authors have declared that no competing interests exist.

through recombination–a process during which a cell infected with two distinct viruses produces "hybrid" viral progeny carrying genetic material from both parents. While recombination is frequent in other RNA viruses, it has not been previously described for the tick-borne encephalitis virus. These findings show that mixture of genetic information from distinct sources can contribute to genetic diversity of this group of viruses, and potentially accelerate their adaptation.

## Introduction

The *Tick-borne encephalitis virus* (TBEV) is a viral species, member of the *Flavivirus* genus, a genus of the *Flaviviridae* family of viruses with relatively short single-stranded positive-sense RNA genomes [1]. The approximately 11000 bp long TBEV mRNA encodes a single polyprotein which is cleaved by both viral and host machinery into three viral capsid forming proteins (C, PrM, E) and seven non-structural proteins (NS1, NS2A, NS2B, NS3, NS4A, NS4B and NS5) [2]. Like many other flaviviruses, TBEV persists in a complex lifecycle of arthropod vectors and mammalian hosts, and human infections are a transmission dead-end [3]. The human form of the infection manifests as mild fever, but in some cases is followed by severe neurological impairments or death [4].

Phylogenetic analysis routinely subdivides all TBEV variants into three distinct subtypes which used to have a strong geographical association. Within Russia, the European subtype was mainly present in the European part but could be encountered all the way to Eastern Siberia; the Far-Eastern subtype occupied the Far East of Russia; while the Siberian subtype was present in the North-Western and Central Russia and predominant in Urals, Western and Eastern Siberia [5], [6]. In recent decades, however, these distribution patterns have become disrupted [7,8] which has been attributed to climate change and increasing human influence on virus migration [9], [10]. Moreover, several novel distinct groups of variants were discovered [11,12]. One of them is the Baikal subtype, which was recently proposed as a novel TBEV subtype candidate [13,14].

Recombination is a major contributor to viral diversity in many studied flaviviruses [15–17]. However, the complexity of TBEV transmission pathways together with a high diversity of hosts has been previously taken to imply that recombination in TBEV is unlikely, and the overall high conservation of TBEV makes it hard to detect if it is present [18]. Nonetheless, as the number of sequenced TBEV variants increased, rare recombination events between different TBEV variants have been proposed [13,19,20]. However, evidence for recombination has been controversial, with each subsequent work disproving previous results while proposing new putative recombination events. The comprehensive study by Bertrand et al. [21] attributed this to the limitations of the viral recombination detection methods when applied to TBEV. Their in-depth simulation established that, to be reliably detectable, the recombinant genomic segments need to be long (>1000bp) or come from distant viral lineages [21].

Therefore, to argue for the presence of recombination in TBEV, one needs to find a variant with a pronounced signal of recombination. Here, we study the Baikal subtype of TBEV. Baikal TBEV variants have been collected for nearly 30 years in large numbers in areas where variants of Siberian and Far-Eastern subtypes are present as well. Currently, there are 22 variants of the Baikal subtype isolated between 1983–1990 in the collection of the Federal State Public Scientific Institution "Scientific Center for Family Health and Human Reproduction Problems" (Irkutsk, Russia) [22]. Additional 6 variants isolated between 1999 and 2010 are in the collection of the Irkutsk Anti-Plague Research Institute of Siberia and Far East (Irkutsk, Russia).

These variants were isolated from *Ixodes persulcatus*, *Myodes rutilus*, *Myodes glareolus* and *Microtus gregalis* collected in Irkutsk region, Buryat Republic and Trans-Baikal Territory of Eastern Siberia. In each of these territories, all TBEV subtypes (European, Siberian and Far-Eastern) coexist and are routinely collected [23]. Mixed infections by TBEV variants of different subtypes, usually Siberian and Far-Eastern, are well described [24]. In particular, TBEV belonging to two distinct subtypes was found in the brain tissue of deceased patients, in the blood samples of ill patients, and in infecting ticks. Notably, among the 10 described individual polytypic samples, 4 were isolated in Irkutsk and Trans-Baikal regions. Among these 4 polytypic samples, 3 contained simultaneously TBEV of Siberian and Far-Eastern subtypes, while the remaining one was a mixture of European and Siberian subtypes [25–27].

Molecular probes show that the Baikal TBEV variants carry fingerprint amino acids unique to Far-Eastern and Siberian subtypes [14]. The unconventional properties of viruses from the Baikal subtype were first discovered in serotyping, when variants of this group showed high antigenic cross-reactivity with variants of Siberian and Far-Eastern subtypes in neutralization tests. Furthermore, the variant 886–84 displayed equivalent affinity to all TBEV subtypes in the agar diffusion precipitation reaction with cross-adsorbed variants-specific serum. All this evidence supports the plausibility of recombination in TBEV in general, and in particular, of recombination involving different subtypes, and makes the Baikal subtype a likely candidate for being recombinant. We hypothesize that the TBEV Baikal subtype has arisen as a result of an ancient recombination event between variants belonging to Siberian and Far-Eastern subtypes.

Many methods for detection of past recombination are available, a number of which are implemented in the RDP package [28]. Most of these methods evaluate the strength of a recombination signal by visualizing the relatedness of the query sequence to other sequences in different regions of the multiple sequence alignment, but ignore the potential diversity of the putative parental variants by uniting them in a consensus sequence, and/or do not implement a formal way to test for statistical difference in relatedness between regions. To address both these shortcomings, we designed a novel approach based on the Grouping Scan analysis (GS) [29]. GS analysis reconstructs the phylogenies in a sliding window along the sequence alignment, and compares the phylogenetic placement of a putative recombinant sequence relative to predefined clades of sequences between different positions of the sliding window. More exactly, for each window, it calculates a score reflecting how deeply the candidate recombinant sequence is embedded into the clades formed by other predefined groups of sequences. Recombination can be inferred if different genomic segments provide conflicting phylogenetic positions for the query sequence, as evidenced by high values of the GS score placing it into different clades. In the present study, we developed a GS-based pipeline that allows us to test for differences in placement of the query sequence between alignment windows, formally assessing the statistical support for recombination. By applying the commonly used methods of the RDP4 package as well as the newly developed GS-based method, we found strong statistically robust support for the hypothesis of the recombinant nature of the TBEV Baikal subtype.

## Materials and methods

### Viral isolation and culturing

Four variants belonging to the Baikal subtype were isolated from ticks (*Ixodes persulcatus*) and small mammals *(Myodes rufocanus)* collected between 1984–1990 in Barguzinsky and Bichursky districts of Buryat Republic, Eastern Siberia. Data on isolation and cultivation of the studied variants are provided in the S1 Table.

## Genome amplification and sequencing

Viral RNA was extracted from 100 μl of cell culture supernatant fluid using a commercial kit Viral RNA mini kit (Qiagen, Germany) according to the manufacturer's instructions. The RNA template was reverse transcribed using the Reverta-L kit (Central Research Institute of Epidemiology, Moscow, Russia). We used RNA isolated from cell culture supernatant fluid for sequencing. The purified PCR products were sequenced bidirectionally using BigDye Terminator v1.1 Cycle Sequencing kit (Thermo Fisher Scientific, Austin, TX, USA) on Applied Biosystems 3500xL Genetic Analyzer (Applied Biosystems, Foster City, CA, USA). The primers used for sequencing are provided in the S2 Table. The sequences were deposited in NCBI GenBank under the following accession numbers: MT708809, MT708810, MT708811, MT708812.

## Alignment preparation

All available TBEV and Omsk hemorrhagic fever virus (OHFV) sequences containing the full protein-coding region were downloaded from the GenBank repository on 11.04.2018. The four Baikal TBEV variants assembled from Sanger sequencing reads were added to this set. Sequence alignment was constructed with Mafft v. 7.408 with default parameters [30]. Sequences with gaps in coding regions and identical sequences were discarded from the alignment. The alignment was trimmed to begin with the main open reading frame [31]. The full alignment contained 11 sequences belonging to the Baikal subtype, 32 sequences of the Siberian subtype, 90 sequences of the Far-Eastern subtype, 47 sequences of the European subtype, and 3 sequences of OHFV. The sequences of OHFV, a different species from the *Flavivirus* genus [1], were added to the alignment to root the reconstructed phylogeny. A maximum likelihood tree was reconstructed on the basis of the whole-genome alignment using MEGA 7 [32] under the GTR+G model with 200 bootstrap trials, and visualized with iTOL [33] (Fig 1).

## Recombination analysis

To search for recombination, we used the three methods available in the RDP4 package [25]: TOPAL, SIMPLOT and BootScan [34–36] under the default parameters, as well as the newly developed method based on the Grouping Scan analysis for recombination (GS) from the Simple Sequencing Editor (SSE) v.1.3 [29]. For this, we assigned all sequences to five groups: OHFV, European, Far-Eastern, Siberian and Baikal. This assignment was performed according to annotation if available, or otherwise according to their grouping into clades with annotated sequences. The few errors in annotations of subtypes were corrected based on the phylogeny in Fig 1. For the methods of the RDP4 package, a consensus sequence was constructed for each group using the simple majority rule. For the GS analysis, a consensus sequence was constructed just for the Baikal subtype, while all sequences from the remaining groups were retained.

To formalize model choice for the GS analysis, we have run the ModelFinder in IQTree2 for each of the 10 random fragments of TBEV sequence alignment, each 250 nucleotides long, matching the window length of GS; as well as the JCF2 and JCF3 fragments (see below). In all 12 tests, the best-fitting model was one of the following: TN, TIM2, TIM3, or GTR. All these models involve more parameters than the most parameter-rich model implemented in GS, i.e., Kimura's 2-parameter model (K80), making K80 our model of choice. Therefore, we used GS with the K80 model. We used sliding window size 250 and step 40. For a comparison of a query sequence and a clade, GS score is defined as, where $y$ is the number of sequences in the group and $N_i$ is the number of nodes separating the query sequence and sequence from the group on the unrooted tree [29]. Low GS scores correspond to high distance between the query sequence and the corresponding clade; GS score of 0.5 corresponds to a sister position

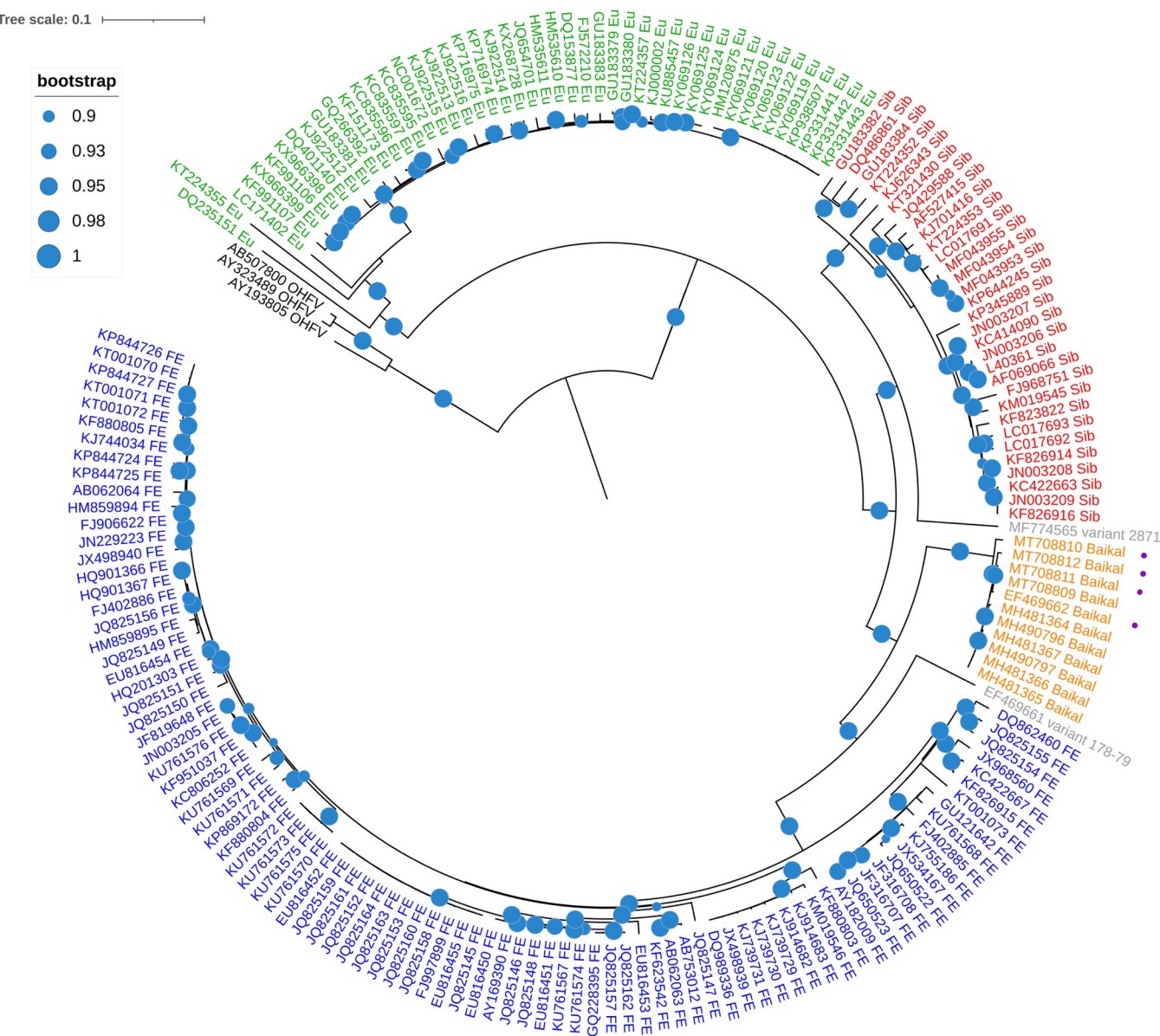

**Fig 1. Reconstructed maximum likelihood whole-genome phylogeny for TBEV variants with 200 bootstrap replicates.** Blue circles indicate bootstrap support of corresponding branches. Leaves are colored by group: red, Siberian subtype of TBEV (Sib); blue, Far-Eastern subtype of TBEV (FE); green, European subtype of TBEV (EU); black, Omsk hemorrhagic fever virus (OHFV) used as the outgroup species; orange, Baikal subtype of TBEV; gray, TBEV variants 2871 and 178–79 excluded from the GS analysis. Newly sequenced Baikal TBEV variants are marked with purple dots.

of a sequence to the clade of interest; and GS score is > 0.5 for a sequence nested within another clade, tending asymptotically to some value < 1 with the increase of the number of nodes separating the sequence from the most recent common ancestor of the clade of interest.

### Estimation of statistical significance of recombination events

For each predefined group of sequences, GS reports grouping score statistics which are calculated for positions of the sliding window along the coordinates of the alignment. We refer to the plot of these values along the alignment as the GS curve. For an alignment window, a GS

score greater than 0.5 for one of the groups indicates that, in the phylogenetic tree reconstructed from this window, the query sequence descends from the most recent common ancestor of this group; in other words, the query sequence is more closely related to sequences from this group than to other sequences. For each window, the set of GS scores for different clades indicates how likely it is to be associated with each of these clades, and the differences in these sets of scores between windows, manifested in rises and falls in the GS curve, suggest that these windows may have different evolutionary histories. However, these scores are not directly interpretable as evidence for recombination or lack thereof.

To address this, we designed a statistic, GS score area under curve ($GSAUC_t$), defined as the total area under the GS curve above some threshold GS score value of $t$. We consider different threshold values of $t$ upwards of 0.5; the value of 0.5 corresponds to the query sequence being the sister to the clade of interest. As the GSAUC statistic characterizes the alignment as a whole, its high values indicate that the query sequence is related to the clade of interest at least in some of the genomic regions. If such elevated relatedness is observed for multiple different clades at distinct alignment regions, resulting in high GSAUC values in multiple comparisons, this suggests that segments of the query sequence differ in their evolutionary history, indicating recombination.

To obtain the null distribution of $GSAUC_t$, we use a permutation procedure, randomly reshuffling all alignment columns in 250 replicates and calculating $GSAUC_t$ for each such reshuffled alignment. While each reshuffled alignment as a whole has the same phylogeny as the original alignment, phylogenies obtained for individual positions of the sliding window generally differ between the original and the reshuffled datasets, resulting in a different value of $GSAUC_t$. The p-value is then calculated as the percentile in the distribution of simulated $GSAUC_t$ values corresponding to the observed $GSAUC_t$. The logic behind this is the following. Reshuffling dissolves all non-independence between neighboring sites, including that caused by similarity of phylogenetic histories. Informally, significant GSAUC implies that genomic segments differ in the degree of phylogenetic affinity they demonstrate towards the considered subtype. While such significance for one of the subtypes may be due to reasons other than the presence of recombination (e.g., differences in conservation between segments), observation of such differences for multiple subtypes has to imply that different genomic windows are differentially affined to different subtypes, implying recombination.

To implement the described procedure, the GS analysis had to be automated. The automation was done on Windows Server 2007 OS with 64 cores and 128 Gb of memory allocated and was controlled by a python script using *pywinauto* package v. 0.6.5 [19]. Up to thirty instances of GS could be launched in parallel under this setting. Python scripts used for calculation of GSAUC and alignments are available at https://github.com/gregoruar/tbev_rec.

## Testing GSAUC on an in silico recombination

To test whether the GSAUC analysis is able to detect ancient recombination events, we used a simulated recombination. For this, we generated an artificial multiple sequence alignment that corresponded to the observed recombination events in the real data. We used ALISIM, an evolutionary sequence simulator that is a part of IqTree 2.2.0 or later versions [37]. ALISIM generates random sequences on the tips of the provided phylogenetic tree using the provided evolutionary model. To model recombination, we first reconstructed the best phylogenies, the best evolutionary models and their parameters for the genome fragments with the strongest signal of recombinations in our data: JCF2, JCF3 and the remaining alignment (non_JCF2_JCF3) (see Results). Using these trees and models, we generated an artificial alignment for each genome fragment.

The positions of the subtypes in the obtained best IqTree trees largely matched the BEAST topologies shown in Fig 4, with the topology of the non_JCF2_JCF3 matching that of JCF1. The inferred evolutionary models for fragments JCF2, JCF3 and non_JCF2_JCF3 were "TIM2e+G4", "TIM2e+I+G4" and "GTR+F+I+I+R3" correspondingly. Finally, we combined these alignments into a complete artificial multiple alignment by inserting the columns of fragment alignments to their original positions in the initial alignment.

We analysed the generated artificial sequence alignment by GS analysis and calculated GSAUC for the GS scores. Both modeled recombination events are detected by GSAUC reliably (S1 Fig).

## Genealogy reconstruction with BEAST

We used the results of the GS analysis to pick the candidate segments of the alignment characteristic of distinct phylogenetic histories (joint characteristic fragments, JCFs). As JCF1, we picked a segment of the alignment representative of the whole-genome phylogeny, namely, the NS1 gene (2329–3388 bp). To choose the segments representative of alternative phylogenies, we reasoned that a single window might exceed the 0.5 GS threshold due to random phylogenetic noise, but this is less likely for multiple adjacent windows. Therefore, for a fragment to be picked as a part of a JCF, we required it to exceed the 0.5 threshold in at least two adjacent windows. There were exactly three such segments in our alignment: the two segments for the Siberian subtype (5520–5770 and 6640–6970 bp, with respectively two and three adjacent windows exceeding the threshold), which we concatenated as JCF2, representing a total of 5.7% of the viral genome; and the single segment for the Far-Eastern subtype (7810–8160, with 8 such adjacent windows; Fig 2D), which we classified as JCF3, representing 3.4% of the viral genome. JCF2 and JCF3 cover the highest GS score peaks in the alignment ($> 0.65$; Fig 2D). Each JCF was then subjected to the analysis of phylogeny (Fig 2D).

For all described parts of the alignment, genealogy was reconstructed by BEAST v.1.10.4 [38]. The same priors were used for each JCF. We performed a formal model selection for BEAST analysis using IQtree [39]. The best-fit model was GTR+F+I+G4 for JCF1, TIM2e+G4 for JCF2, and TIM+F+I+G4 for JCF3. For all these models, the closest model among those implemented in BEAST was GTR+Gamma, which we used. Uncorrelated lognormal clocks were modeled as lognormal distributions of substitution rates among lineages with the mean of $10^{-4}$. The prior for the age of the root was modeled by normal distribution with a mean of 5000 years and standard deviation of 1000 years. Calibrated Yule model with default parameters was used as the prior on the tree shape best representing the heterogeneity of the dataset. The phylogeny of JCF2 was reconstructed in a single run with the substitution model and mutation rate being unlinked. Each BEAST analysis was performed for 50 million generations with 10% of samples discarded at burn-in. All BEAST runs were checked for convergence by controlling for effective sampling size (ESS values $> 200$). Visualization was done in ggtree R package [40].

## Results

### The Baikal subtype is evident of recombination

We characterized the genetic diversity of TBEV by phylogenetic analysis. The obtained TBEV phylogeny (Fig 1) consists of the three main genetic groups with high bootstrap support corresponding to the accepted classification of the virus into three subtypes: European, Siberian and Far-Eastern [5]. Our four newly sequenced isolates cluster together with other variants belonging to the Baikal subtype, forming a well-supported sister clade to the Far-Eastern subtype. Similar to other studies [41], the variants we obtained underwent several rounds of mouse and

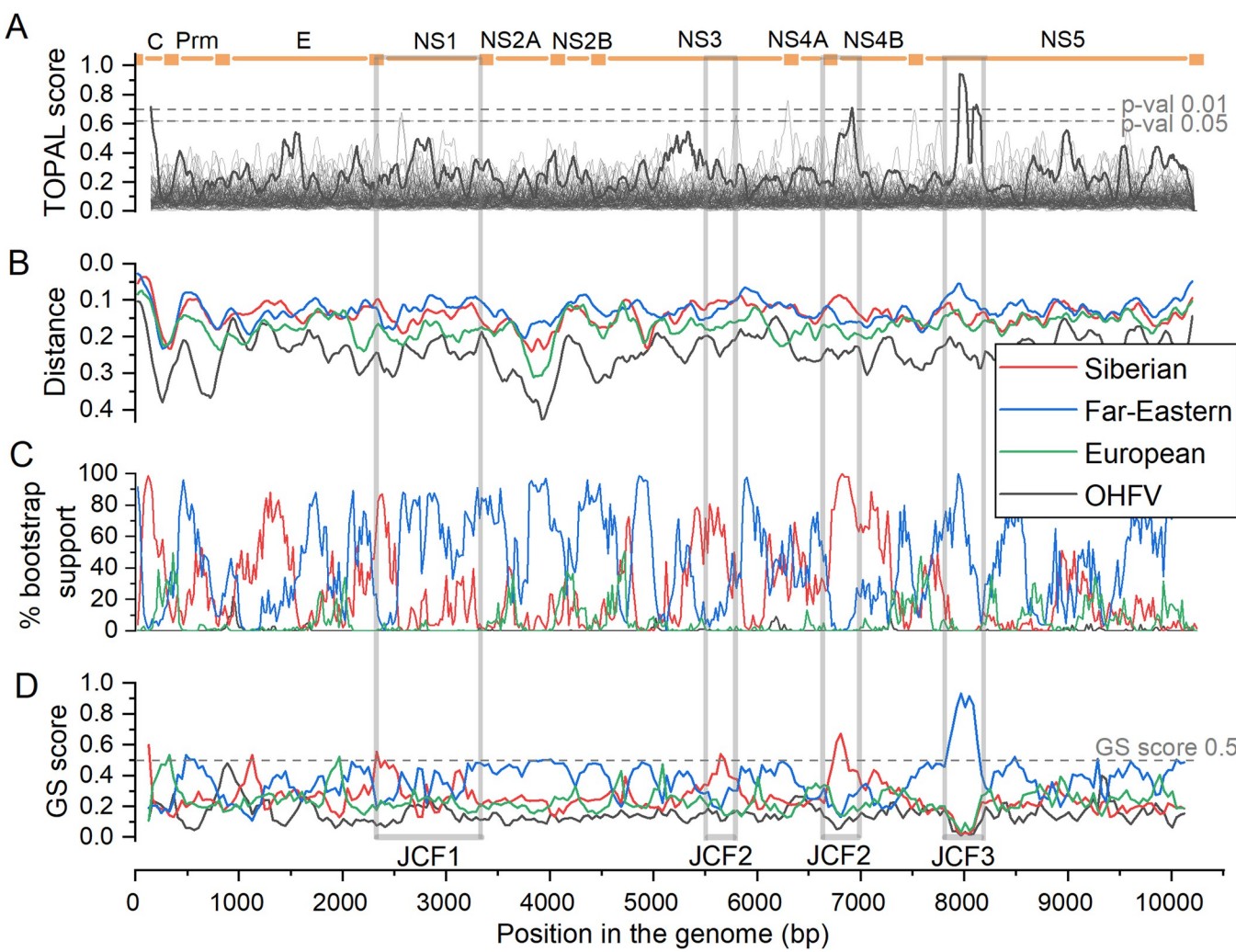

**Fig 2. Evidence for recombination at origin of the Baikal subtype.** Each analysis compared five consensus sequences: of the Baikal subtype (used as the query), and of the four other subtypes. (A) TOPAL/DSS score, reflecting the difference in shapes of the phylogenies between the adjacent alignment windows (dark gray), against the background of the same values calculated for bootstrap replicates of simulated non-recombinant sequences (light gray); the two dashed lines indicate the expected 99% and 95% confidence intervals of expected scores [36]. (B) SIMPLOT score, reflecting the nucleotide sequence distance between the query and the four consensus sequences [35]; the Far-Eastern subtype is the closest to the Baikal subtype throughout most of the alignment, but the Siberian subtype is more closely related in some of the alignment regions. (C) Bootscan plot is based on reconstructing the neighbor joining trees for each sequence window, and calculating the bootstrap support for the clade uniting the query subtype and each of the four other subtypes [34]; again, the clade uniting the Baikal subtype with the Far-Eastern subtype has the highest support for the bulk of the alignment, but the clade uniting it with the Siberian subtype has a higher support for some regions. (D) GS analysis. Joint characteristic fragments (JCFs) used for subsequent phylogenetic reconstruction are shown below the GS scores plot. The 0.5 threshold is shown as a dashed gray line. For (A)-(C), default parameters were used, namely, sliding window of length 200, step 20 (for (A), step 10 + smoothing step 10). For (D), we used the sliding window length of 250, and step size of 40.

tissue culture propagation, potentially introducing new mutations; however, the bulk of the divergence of this clade was in its ancestral branch (Fig 1), indicating that the contribution of such mutations was negligible if present at all. Overall, the obtained whole-genome phylogeny conforms with the previous results [21]. variants TBEV-178-79 (GenBank ID EF469661) and TBEV-2871 (GenBank ID MF774565) were found to be remote outgroups to the Far-Eastern and Siberian subtypes respectively (Fig 1). variant 178–79 is hypothesized to be the sole representative of a distinct TBEV subtype [42], and variant 2871 was acknowledged to form a new lineage in Siberian TBEV subtype [11]. As these variants are sole representatives of their clades,

we have no way to exclude cross-contamination or additional recombination events involving them; therefore, we excluded them from subsequent analysis.

To formally ask whether the Baikal subtype descends from a recombination event, we used a series of tests. First, we applied three alignment-based tests available in the RDP4 package: TOPAL, SIMPLOT and BootScan to consensus sequences of the five studied groups. All three tests support recombination between the Far Eastern and the Siberian groups at the origin of the 886-like group (Fig 2). Specifically, both SIMPLOT (Fig 2B) and BootScan (Fig 2C) indicate that while the Baikal subtype is more closely related to the Far-Eastern subtype in most alignment regions, it is more closely related to the Siberian subtype in some others, notably, around position 6750. Besides, around position 8000, the Baikal subtype clusters with the Far-Eastern subtype more reliably, and at a lower sequence distance, than in surrounding positions. Both the vicinity of position 6750 and of position 8000 demonstrate an abrupt change in the shapes of the phylogenies between the adjacent positions, as evidenced by high TOPAL/DSS scores (Fig 2A).

Second, to better understand the history of recombination events at the origin of the Baikal subtype, we used the GS analysis. GS revealed a signal of recombination for the Baikal subtype (Fig 2D): for some segments of the alignment, GS score was above 0.5 for Siberian subtypes, while for others, it was above 0.5 for Far-Eastern subtypes, suggesting that it has descended from a recombination event involving these two subtypes.

## Statistical support for recombination

The results of the GS analysis indicate that different segments of the 866-84-like genome are nested within different subtypes, suggesting recombination. To formally test this, for each of the four predefined groups (European, Far-Eastern, Siberian TBEV subtypes and OHFV), we tested the hypothesis that the 866-84-like clade is equally deeply embedded within the considered subtype across all genomic windows. If this hypothesis is rejected for more than one subtype, this would imply that different segments of the 866-84-like genomes are more closely related to different subtypes.

As the test statistic, we calculated the GSAUCs for the consensus sequence of the Baikal TBEV clade. High values of $GSAUC_t$ for a threshold $t > 0.5$ imply a high degree of relatedness, for some genomic segments, between the query sequence and the clade of interest. A comparison with a single clade can result in significant $GSAUC_t$ values for chance reasons such as unevenness of the substitution rate (e.g. due to differences in mutation rate or selective convariantt) across the query genome. However, such unevenness cannot result in unexpectedly high $GSAUC_t$ values in comparisons with multiple different clades. Instead, such a pattern has to imply that segments of the query sequence differ in their relatedness to different clades, implying recombination.

To statistically assess the increase in a $GSAUC_t$ value, we calculated the p-values by comparing the actual $GSAUC_t$ value to its null-distribution (see Methods). In order to explore the whole GS score curve, we performed this calculation for different GS thresholds above 0.5.

Significantly ($p < 0.05$) elevated $GSAUC_t$ values are observed in the comparisons of the 866-84-like sequence with the Far-Eastern and the Siberian TBEV subtypes (Fig 3). The significance in these comparisons depends on the threshold chosen. Generally, the ranges of the threshold values for which $GSAUC_t$ is significant reflect the ranges of GS scores that are outlying compared to the baseline in Fig 2D. On the one hand, for the Far-Eastern subtype, we observe low p-values (down to 0.0) for the threshold values ranging between ~0.72 and ~0.93, indicating elevated $GSAUC_t$ values for these thresholds. These thresholds correspond to the elevated GS score in JCF3 (Fig 2D). On the other hand, for the Siberian subtype, we observe a

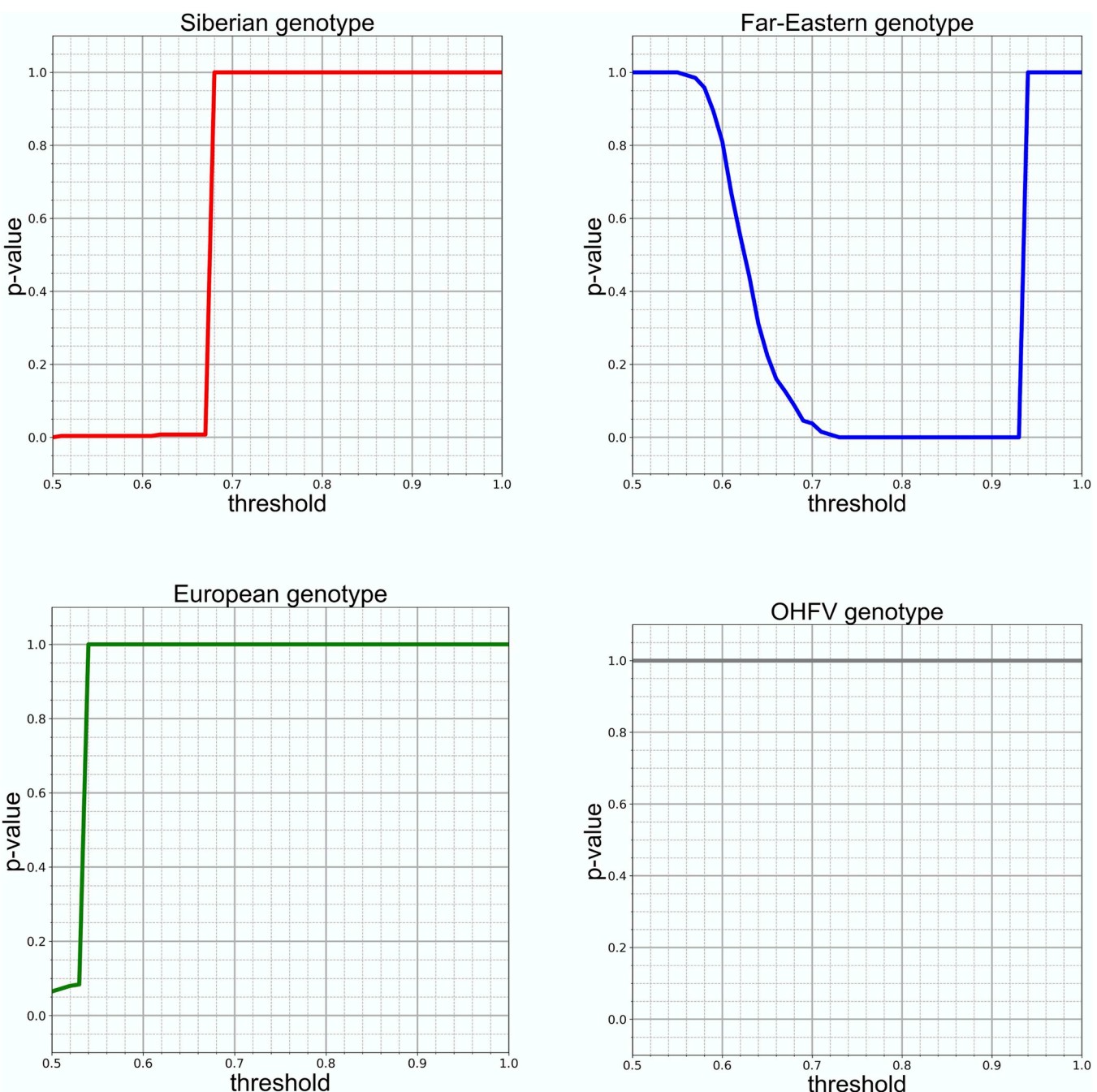

**Fig 3. Phylogenetic evidence for recombinant origin of the Baikal subtype.** The plots show the dependence of the p-value for the GSAUC statistic on the GS score threshold (with threshold increments of 0.01), relative to each of the four TBEV subtypes. While no GS score thresholds result in p<0.05 for the European subtype and the OHFV (an outgroup species), for the Far-Eastern and Siberian subtypes, some of the GS score thresholds correspond to p<0.05, indicating that both these subtypes contributed to the Baikal subtype.

low p-value (down to 0.0) for threshold values up to ~0.67, corresponding to the elevated GS score in JCF2 (Fig 2D). The non-monotonic dependence of p-values on the threshold is unsurprising: a p-value of 1 is expected both for low thresholds (as the baseline GS score is rather uniform across the genome) and for high thresholds (as the GS score never reaches values above ~0.95, Fig 2D). For the European subtype, the GS scores are somewhat elevated for low

threshold values (<0.55); however, this decrease is not statistically significant (p = 0.069), and the fact that it is only observed for threshold values close to 0.5 suggests that it is due to noise. The OHFV group has high p-value for all threshold values, as expected from the outgroup subtype.

## Evolutionary history of the Baikal subtype

To study the origin of the putative recombinant fragments identified with the GS analysis, we reconstructed their genealogy using BEAST. For this analysis, we selected three groups of alignment columns (joint characteristic fragments, JCFs) each corresponding to one of the three patterns in the GS analysis: low GS scores for all comparisons (positions 2329–3338, JCF1); high GS scores for grouping with the Siberian subtype (concatenated positions 5520–5770 and 6640–6970, JCF2); and high GS scores for grouping with the Far-Eastern subtype (positions 7810–8160, JCF3). The resulting phylogeny (Fig 4) contains posterior probability (PP) of the nodes that determine the phylogenetic position of the Baikal subtype; PP and heights of other nodes are available in S2 Fig and S3 Table.

The genealogy of the non-recombinant JCF1 has the same structure as the ML tree for the whole genome: the Baikal subtype branches off from the most recent ancestor of the Far-Eastern subtype, being the sister group to all Far-Eastern sequences. This topology is supported by the high posterior probability (PP) of three nodes: uniting the Baikal subtype into a clade (PP > 0.99, node I of S2 Fig), uniting the Far Eastern sequences into a clade (PP > 0.99, node F), and indicating that these two clades are in a sister relationship (PP > 0.99, node C).

By contrast, in the phylogenetic tree reconstructed from JCF2, the Baikal subtype is the sister group to all Siberian sequences. Again, this relationship is supported robustly by the high PP of three nodes: uniting the Baikal subtype into a clade (PP > 0.99, node I), uniting the Siberian sequences into a clade (PP > 0.99, node E), and indicating that these are sister clades (PP > 0.99, node D).

Finally, in the phylogeny of JCF3, the Baikal subtype is embedded in the Far-Eastern subtype. The clustering of all Baikal TBEV sequences into a group and of the Far Eastern sequences into a group are each supported by high PP (PP > 0.99 (node I) and PP > 0.99 (node J), respectively); however, the Baikal TBEV clade is nested within the Far Eastern clade, although the support for this nesting is lower (PP = 0.61 (node G) and PP = 0.77 (node H) for two successive nodes).

## Discussion

The probability of a recombination event and the ease with which it can be detected both depend on the evolutionary distance between the parental lineages. Recombination between closely related variants is expected to be more frequent as such variants are more likely to circulate at the same place and time, exchange of genetic segments is mechanistically easier, and is less likely to incur a fitness cost due to epistatic interactions between sites; but it is harder to detect as its footprint on sequences is weaker [21]. All methods for detection of recombination, including our newly developed GSAUC statistic, are expected to have more power for more distantly related sequences. Most previous effort to detect recombination in TBEV has focused on within-subtype recombination, and evidence for it has been controversial [13,18–21]. Here, we address a putative recombination event involving two major TBEV subtypes: Siberian and Far Eastern, and provide evidence that at least one recombination event between these subtypes has been at the origin of the Baikal subtype of TBEV.

These findings have the following limitations. First, our inference of recombination is based on an incongruence of phylogenetic trees for different alignment regions. While

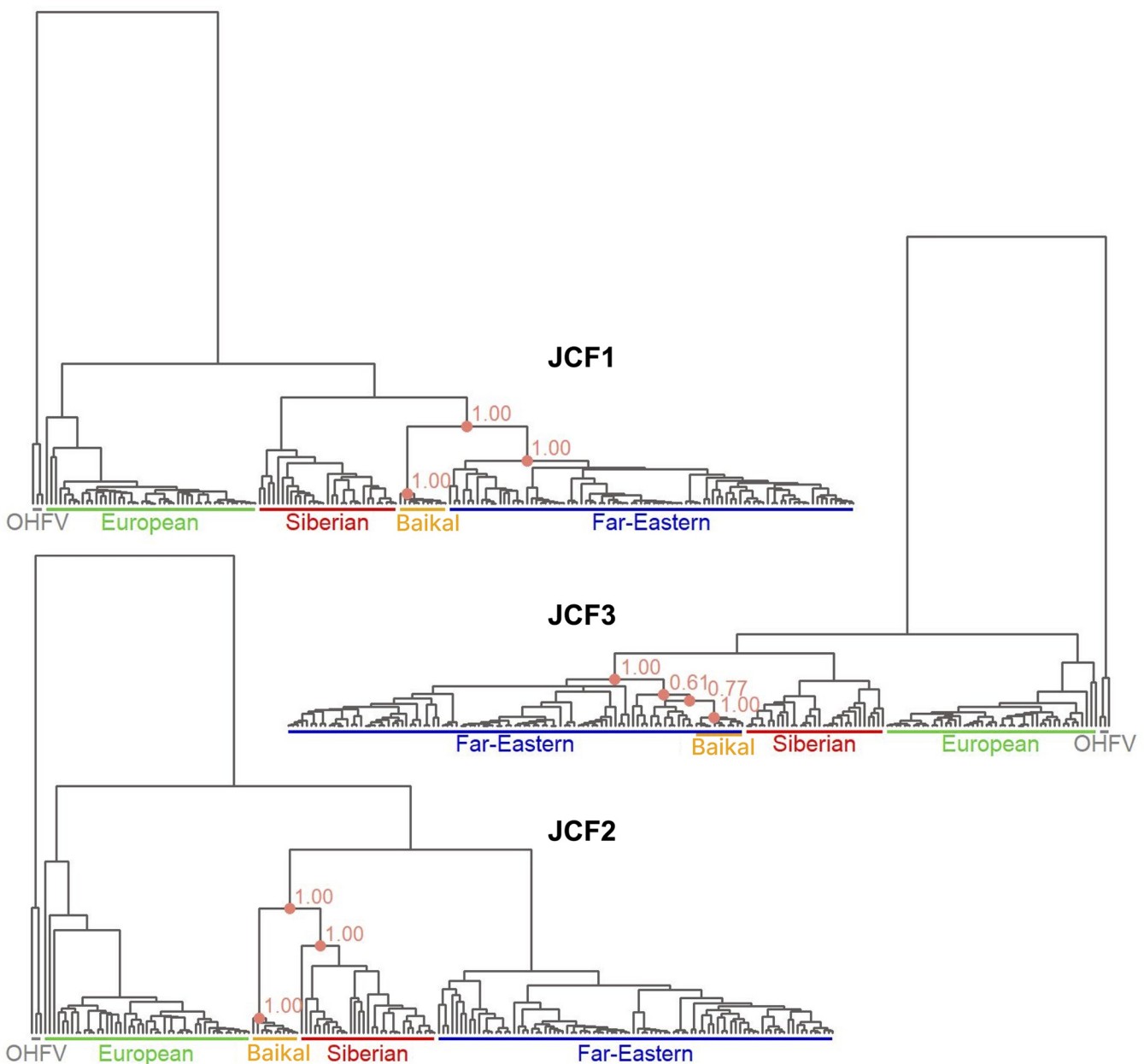

**Fig 4. Reconstructed Bayesian phylogenies of TBEV based on the characteristic fragments.** Each panel is based on the alignment segment of the corresponding JCF.

homologous recombination is the most likely source of such incongruence, other factors can also contribute to it, including non-uniform substitution rates between sequence regions and convergent evolution caused by biased mutation patterns or the action of natural selection. Such convergence would have to affect multiple sites, both synonymous and nonsynonymous, making this possibility unlikely; still, it is possible. Second, our analysis is based on just 11 samples of the Baikal subtype of TBEV. Including additional samples may reveal a more complicated history, including additional recombination events. This can be addressed by more extensive sampling of the Baikal subtype as well as other related subtypes. Third, the current

methods for recombination detection are unable to precisely pinpoint the recombination breakpoint, particularly if the recombination was old. Fourth, the temporal signal of the inferred recombination events is weak, complicating their dating (see below).

Given these limitations, our claim for recombination at origin of the Baikal subtype is based on two types of phylogenetic evidence, both with robust statistical support. First, using existing methods as well as a newly developed GSAUC statistic, we show that fragments of the TBEV genome differ in which other clade they are more similar to. The methods used to show this are based on different approaches, indicating that the support for recombination is unlikely to be a methodological artifact. Second, using Bayesian phylogenetics, we show that they respectively cluster with different TBEV subtypes.

The GS plot indicates that the segments involved in recombination are relatively short (Fig 2D). Therefore, it is critical to integrate its signal over sequence segments. The developed GSAUC statistic provides a framework for such integration. Similar to the original GS analysis which underlies it, it is based on the idea that, unless the recombination events were too frequent, adjacent genomic positions tend to share phylogenetic history. Therefore, even if the signal is partially eroded by subsequent evolution, we would expect higher cumulative GS scores relative to some predefined clades than would be expected for a reshuffled genome sequence.

The GS plots allow to single out putative regions with different evolutionary histories, JCF1, JCF2 and JCF3, which can then be analyzed phylogenetically. While other regions could have also been involved in recombination, these are the genomic segments for which the evidence for recombination is the most robust (Fig 2D).

Remarkably, the reconstructed phylogeny for the JCF1, JCF2 and JCF3 strongly supports the hypothesis that the Baikal subtype originated through recombination. In particular, the tree of JCF1 features the Baikal subtype as an outgroup for the Far-Eastern subtype with very high posterior support, while the tree of JCF2 strongly backs the placement of this group with the Siberian subtype. The discrepancy between these trees implies recombination between the Siberian and Far-Eastern subtypes of TBEV at the origin of the Baikal subtype. The tree of JCF3, the genomic region with very high GS similarity to the Far-Eastern subtype, shows deep embedding of the Baikal TBEV clade into the Far-Eastern subtype, although with moderate posterior support. The difference between the JCF1- and JCF3-based phylogenies may indicate the occurrence of an additional more recent recombination between the ancestor of the Baikal subtype and Far-Eastern subtype.

When did these events take place? In theory, evolutionary events can be dated using a molecular clock analysis. Such inference is complicated by the apparent weakness of the temporal signal in the TBEV genetic data [43,44]. For example, Deviatkin et al. [43] have not observed positive correlation between sampling date and root-to-tip distance for divergence at the scale above major TBEV subtypes, although such correlation has been observed within most subtypes (Far Eastern, European and two out of three Siberian subclades). Moreover, even in the presence of a temporal signal, precise dating of evolutionary events based on three phylogenetic trees with different topology and evolutionary rates is challenging. With these caveats in mind, we attempt to roughly date the major events in the evolutionary history of the Baikal TBEV clade. To cross-reference the branching events in different trees, we use the fact that the last common ancestor (LCA) of all considered TBEV subtypes should coincide between them. Indeed, its timing is similar (3731–4546 y.a.) between all three trees (S2 Fig).

Comparison of the phylogenies constructed from JCF1, JCF2 and JCF3 suggests the following timeline (S2 Fig). The Far-Eastern and Siberian subtypes diverged approximately 676–1434 y.a. The LCA of the Far-Eastern subtype and the Baikal TBEV clade date to 688 y.a., according to the JCF1 tree; and the LCA of the Siberian subtype and the Baikal TBEV clade

date to 973 y.a., according to the JCF2 tree. This implies that the first recombination event involving the Far-Eastern and Siberian subtypes dated in this time interval. According to the JCF3 tree, the LCA of the Far-Eastern subtype and the Baikal TBEV clade dates to 235 y.a., implying that the second putative recombination event involving the Baikal TBEV clade and the Shenjang lineage within Far-Eastern subtype dated between 688 and 235 y.a. The LCA of all known representatives of the Baikal TBEV clade dates to 83–119 y.a. The 95% HPDs on these dates are very high (S2 Fig), and therefore they should only be considered rough estimates.

In conclusion, we detect an instance of recombination in the history of the TBEV Baikal subtype. This inference is supported by multiple methods, including a newly developed GSAUC method based on the GS statistic which allows to measure the signal of recombination in a statistically rigorous way. This discovery concludes a long debate regarding the possibility of recombination in the evolution of TBEV. The role of such events in the origin of other TBEV variants remains an important subject for future studies.

## Supporting information

**S1 Table. Isolation and cultivation of the studied strains.**
(PDF)

**S2 Table. Primers used for sequencing.**
(PDF)

**S3 Table. Heights and posterior probabilities of select nodes shown in S2 Fig from the reconstructed Bayesian phylogeny of TBEV JCFs.**
(PDF)

**S1 Fig. GSAUC reliably detects simulated recombination event.** (A) GS analysis of the artificially generated alignment with recombination; (B) GSAUC curves corresponding to the GS analysis.
(TIF)

**S2 Fig. Nodes from the reconstructed Bayesian phylogeny of TBEV JCFs for which heights and posterior probabilities are presented in S3 Table.**
(TIF)

## Author Contributions

**Conceptualization:** Georgii A. Bazykin, Alexey D. Neverov.

**Data curation:** Grigorii A. Sukhorukov.

**Formal analysis:** Grigorii A. Sukhorukov, Alexey D. Neverov.

**Funding acquisition:** Georgii A. Bazykin.

**Investigation:** Grigorii A. Sukhorukov, Alexey I. Paramonov, Oksana V. Lisak, Irina V. Kozlova, Georgii A. Bazykin, Alexey D. Neverov, Lyudmila S. Karan.

**Methodology:** Grigorii A. Sukhorukov, Georgii A. Bazykin, Alexey D. Neverov.

**Project administration:** Georgii A. Bazykin.

**Resources:** Alexey I. Paramonov, Oksana V. Lisak, Irina V. Kozlova, Lyudmila S. Karan.

**Software:** Grigorii A. Sukhorukov.

**Supervision:** Georgii A. Bazykin, Alexey D. Neverov.

**Visualization:** Grigorii A. Sukhorukov.

**Writing – original draft:** Grigorii A. Sukhorukov.

**Writing – review & editing:** Georgii A. Bazykin, Alexey D. Neverov, Lyudmila S. Karan.

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
