## [Decision Letter · Decision Letter 0]

17 Jul 2022

Dear Dr. Bazykin,

Thank you very much for submitting your manuscript "The recently identified long-living lineage of the tick-borne encephalitis virus is a result of an ancient recombination event" for consideration at PLOS Neglected Tropical Diseases. As with all papers reviewed by the journal, your manuscript was reviewed by members of the editorial board and by several independent reviewers. In light of the reviews (below this email), we would like to invite the resubmission of a significantly-revised version that takes into account the reviewers' comments. 

please revise the manuscript based on the formatting requirements, you can find it at https://journals.plos.org/plosntds/s/submission-guidelines

We cannot make any decision about publication until we have seen the revised manuscript and your response to the reviewers' comments. Your revised manuscript is also likely to be sent to reviewers for further evaluation.

Sincerely,

Wen-Ping Guo

Academic Editor

Samuel Scarpino

Section Editor

please revise the manuscript based on the formatting requirements, you can find it at https://journals.plos.org/plosntds/s/submission-guidelines

Reviewer's Responses to Questions

**Key Review Criteria Required for Acceptance?**

**Methods**

-Are the objectives of the study clearly articulated with a clear testable hypothesis stated?

-Is the study design appropriate to address the stated objectives?

-Is the population clearly described and appropriate for the hypothesis being tested?

-Is the sample size sufficient to ensure adequate power to address the hypothesis being tested?

-Were correct statistical analysis used to support conclusions?

-Are there concerns about ethical or regulatory requirements being met?

Reviewer #1: Revisions ...

Pg. 6 Par. 2 Line 1: How did the authors determine which model (i.e., Kimura’s 2-parameter model) best fit the applied TBEV sequence alignment?

Comments & Suggestions ...

Methods: Inclusion of an additional positive control or engineered in silico model for TBEV recombination (similar to the cited Bertrand et al. reference) may further strengthen the robustness of the proposed GSAUC method. Does this method detect artificially-introduced recombination events? Furthermore, how do the alternative RDP4 methodologies perform as a baseline in these same scenarios?

Reviewer #2: - The authors excluded two most divergent variants of the Siberian and Far-East genotypes (TBEV-178-79 (GenBank ID EF469661) and TBEV-2871 (GenBank ID MF774565)) from the presented analysis. Why not improve the power of the analysis, given the otherwise limited virus sampling and low genomic variation that hindered prior analyses? If these viruses are indeed recombinants, as the authors concern, any evidence in support of this assertion will strengthen the main conclusion of the study. If the inclusion of these two sequences affects the main conclusion of this study, this result could be used to reveal critical dependencies of the recombination analysis and call for further expansion of genomic exploration of the TBEV natural diversity. One way or another, confidence of conclusions, original or revised, would be improved. 

- Consensus sequence versus all sequences. The authors used consensus sequences of genotypes (groups) for recombination analysis without providing a rational and acknowledging that it may decrease the analysis power. Besides, there are two other issues with this choice: a) no details provided about how consensus sequences were generated, including the use of weighting or another method to correct for uneven virus sampling (even if the two most divergent sequences were excluded); b) despite consensus sequence being used for each group, GS score calculation relies on analysis of all sequences in each group (p. 6), which is confusing.

- Provide a Supplementary Table detailing sequences used in the study

Reviewer #3: Objectives of the study are clearly articulated, the design is appropriate.

**Results**

-Does the analysis presented match the analysis plan?

-Are the results clearly and completely presented?

-Are the figures (Tables, Images) of sufficient quality for clarity?

Reviewer #1: Revisions ...

Pg. 13 Par. 3: Several indicated node and LCA values for the JCF1 and JCF2 phylogenies (e.g., nodes C and D) are inconsistent with the referenced supplemental Figure S1 infographics.

Figure S1: Based on the JCF2 phylogeny, the associated metadata for Node C appears to be incorrectly listed in the JCF1 versus intended JCF2 column. In addition, similar metadata for Node D is listed in the JCF2 column despite annotation on the JCF1 phylogeny. Relative to Figure 5, these discrepancies may stem from the apparent swap of the labeled JCF1 and JCF2 phylogenies in Figure S1.

Comments & Suggestions ...

Pg. 9 Par. 2 Line 3-5 (and Figure 2): Expand upon the provided text to clarify how the results of the applied RDP4 package analytics: TOPAL, SIMPLOT, and BootScan support signatures of Far Eastern and Siberian genotype recombination. Inclusion of these tests provides an analytical foundation the authors build upon in the proposed GSAUC method, and readers may be unfamiliar with interpretation of these results. In brief, what particular features of the TOPAL, SIMPLOT, and BootScan output support (or potentially contradict) the results of the applied GSAUC method? Minimally, expansion of the descriptive text in the Figure 2 legend would be highly informative (such as the labeled thresholds and gray-scaled line series in the TOPAL/DSS score panel).

Figure 3: Based on the provided inclusion criteria and the results detailed in Figures 3 and 4, GS scores for the Siberian genotype exceeded the defined 0.5 threshold with GSAUC statistic p-values < 0.05 in the encoded capsid and prM-E gene segments. Why did the authors disqualify these signals as putative joint characteristic fragments (JCFs)?

Reviewer #2: - Provide a chapter describing four new TBEVs and their genome sequences and relationship with known TBEVs. Acknowledge the mouse and tissue culture propagation of TBEV as sources of extra mutations. Its (limited?) scale and impact on downstream inferences should be discussed in the Discussion section. 

- p. 9: detail results presented in Fig. 2, e,g. plot distribution, peaks, agreement between different methods, type of evidence, etc. 

- the time-based analysis should be moved from Discussion p.13 to the Results if retained in the paper at all, which is uncertain. The lack of the root-to-tip regression signal and high uncertainty of the HPD ranges (see below) question reliability of this analysis. Choice of the outgroup and testing of evolutionary models should be substantiated. 

- Move Fig. 3 GS and genomic plots to the bottom and top of Fig. 2, respectively. This will facilitate comparison and reveal agreement between results of different methods. 

- Figure 4: explain why plots look continuous rather than discrete functions, despite the underlying analysis uses a sliding window with a discrete step. Could the function be extended below 0.5? Could a log scale be used at the Y axis and the p-value cut-offs of 0.05 and 0.01 be specified ?

- What is a tree topology of phylogeny that is based on the entire genome minus JCF2 and JCF3? Does it match the topology of Fig. 1 and JCF1 trees? If so, why not use it instead of JCF1, which selection is a bit arbitrary?

- Provide sequence alignment(s) to illustrate and support recombination

Reviewer #3: The results are clearly and completely presented.

**Conclusions**

-Are the conclusions supported by the data presented?

-Are the limitations of analysis clearly described?

-Do the authors discuss how these data can be helpful to advance our understanding of the topic under study?

-Is public health relevance addressed?

Reviewer #1: Comments & Suggestions ...

Discussion: Sukhorukov et al. might consider addressing how this new methodology can be applied more broadly in their field. Is the GSAUC method limited to the detection of ancestral recombination events (due to underlying model assumptions/parameters), or can this tool be applied to more recent evolutionary scenarios in other viral systems? Inclusion of additional text would highlight inappropriate applications and strengthen readership impact/utility.

Reviewer #2: - See Methods

- The manuscript is dominated by the use of “recombination” without acknowledgement that obtained results are about the incongruence of trees for different genomic regions, with a homologous recombination being one of possible explanations of the observed patterns. Accordingly, the word “recombination” should be used carefully and seldom, and other explanations for the obtained results should be presented and, if discarded, explicit reasoning should be provided. - The same is implied to the title and conclusions. 

- Limitations of the study are yet to be specified

Reviewer #3: The conclusions are correct and supported by the data presented.

**Editorial and Data Presentation Modifications?**

Reviewer #1: Revisions ...

Pg. 4 Par. 2 Line 17: Provide the expanded definition for “AUC” in the “GSAUC” acronym.

Figure 1: Consider editing the applied color-scheme for the bootstrap legend to allow better visibility/step differentiation (such as an orange-blue scale). Currently, it is difficult (partially due to size) to discriminate between the green lines (0.95 and 1.00) and find the bright yellow (0.90) colored lines.

Comments & Suggestions ...

Figure/Table Descriptions: In general, provision of more descriptive text for applied data series, axes, thresholds, color-schemes, icons, and numerical labels will aid in orienting readers and data interpretation.

Reviewer #2: Introduction. 

- Note that most analysed TBEV sequences are just sequences, they are “variants” but not “strains”, that designation is reserved for well characterized variants.

- Up-to-date taxonomy of the Flaviviridae including how it must be spelled and written is here (10.1099/jgv.0.000672)

- The presented analysis concerns a flavivirus species, which includes all variants of TBEV and this species should be specified. Consult https://doi.org/10.1371/journal.ppat.1009318 for understanding differences between viruses and virus species. 

- TBEV genome is mRNA that is translated to produce polyprotein, cleaved by virus and host proteases to mature proteins.

- Describe the basis of the TBEV genotyping, e.g. genome region, method, criterion etc 

- Describe evidence for homologous recombination in TBEV (genome regions, genotypes) and causes of uncertainty of these findings.

- p.3. Provide reference for the observation of infection by multiple TBEV genotypes of a patient. 

- “In the agar…” sentence does not have sense

Discussion

- p.11: discuss “evidence for it [recombination] has been controversial”. Show how the controversial points were addressed in this study, especially a role of the new statistics; why it is superior compared for the Topal statistic, for instance.

- p.11: make explicit reference to tree nodes (A, B, C, etc) in Fig. S1, when discussing the results.

- p.12: based on the Results section, it is misstatement to list the JCF1 region as of the possible recombinant origin; should be corrected 

Others:

- Specify genome fractions occupied by JCF2 and JCF3

- Could you say that JCF2 and JCF3 are the only regions of possible recombinant origins, given the analyzed dataset?

- Illustrations and elsewhere: make clear difference between TBEV genotypes of the same virus species and the outgroup, which belongs to another flavivirus species.

- Figures 2-4: specify query

- Fig. S1: a) Results for JCF1 and JCF2 may have been swapped relative to Fig. 5 and Fig. S1-associated Table; b) 95% HPD for many nodes are highly uncertain that need to be discussed in respect to the time-based inferences; c) some nodes for three phylogenies must have different names (e.g. F node for JCF3 versus JCF1 and JCF2)

Reviewer #3: Specific comments: 

1. Title: It would be good to change the title to reflect better the obtained results. For example, the Baikal TBEV genotype is a result of a recombination between Siberan and Far-Eastern TBEV strains.

2. Abstract: The Baikal and Himalayan genotypes are already considered as new TBEV genotypes. 

3. Abstract: TBEV samples – better: TBEV strains

4. Introduction: Flavivirus genus is not the only genus of the family Flaviviridae

5. The reference number 1 is not appropriate. Please, cite any recent review or ICTV release. 

6. “short single-stranded positive-sense RNA viruses” – unusual wording

7. “encodes a single transcript that is translated” – better: encodes a single open reading frame that is translated

8. it is not clear how the non-structural proteins are involved in transmission of the virus?

9. Introduction second paragraph: Far East (capital F)

10. Page 3 and further: in the scientific literature a name “Baikalian subtype/genotype” is already well established instead of “886-84-like”. The “886-84-like” name is not well known in the Western literature and most papers use just the name “Baikalian”. This reviewer strongly encourages to use the name Baikalian genotype thorough the manuscript. 

11. Page 3 end of the second paragraph: a reference is missing

12. Page 3 third paragraph: not “reaction of neutralization” but “neutralization test” or “neutralization assay”

13. Page 4 second paragraph: … study, we developed….

14. Figure 2: please, prepare a better figure legend. It is not good to say only “results from the analysis”. The legend should be self-explanatory; i.e., describe what was the purpose, what was done and what is shown. The same for Figure 2 and 5.

**Summary and General Comments**

Reviewer #1: In this manuscript, Sukhorukov et al. introduce a newly developed method to infer ancestral recombination events in the tick-borne encephalitis virus (TBEV) genome. Here, the authors implemented a recombination model (i.e., GSAUC method) to specifically detect and validate evolutionary signals of recombination events within the TBEV 886-84-like lineage; in particular, the evidence for two proposed recombination signatures were outlined between the ancestral Far Eastern (FE) and Siberian genotypes and within the more recent Far Eastern Shenjang lineage and 886-84-like clade.

This document is well-written. In particular, the implemented GSAUC method computational model, hypothesis testing, and data interpretation/validation are all outlined in an intuitive framework which will allow readers to reproduce or translate the proposed method via the provided GitHub link and referenced software packages. In preparation, Sukhorukov et al. could consider further validation of the robustness of their testing algorithm using engineered in silico data models or additional, comparative software packages/code repositories.

Reviewer #2: Sukhorukov et al., present new genome sequences of four TBEV collected in the Baikal area of Russia over six years and phylogenomic analysis of TBEV sequences available on 2018 along with the newly sequenced genomes. Using diverse software and an original statistical test, developed for this study, they found support for conflicting evolutionary histories of different genome regions encoded by a TBEV group known as 886-84-like cluster (genotype), including the new genome sequences. The authors proposed that an ancestor of this cluster emerged as result of ancient recombination(s) between ancestors of two main TBEV genotypes, Siberian and Far-East. 

The conducted analysis is informative (after addressing the criticisms), although hardly definitive in resolving the uncertainty concerning a role of homologous recombination in the evolution of TBEV, contrary to the authors claim.

Reviewer #3: Sukhorukov et al. submitted a manuscript titled “The recently identified long-living lineage of the tick-borne encephalitis virus is a result of an ancient recombination event” for peer-review procedure in PLoS Neglected Tropical Diseases. The manuscript identified that the newly described Baikalian TBEV subtype may be a result or a recombination event between Siberian and Far-Eastern TBEV strains. This discovery is of interest and provide the first clear and statistically solid evidence of recombination in TBEV. However, the title and several other issues listed above need to be addressed before the manuscript can be considered acceptable for publication.

PLOS authors have the option to publish the peer review history of their article (what does this mean?). If published, this will include your full peer review and any attached files.

Reviewer #1: No

Reviewer #2: No

Reviewer #3: No
---

## [Decision Letter · Decision Letter 1]

20 Nov 2022

Dear Dr. Bazykin,

Thank you very much for submitting your manuscript "The Baikal subtype of tick-borne encephalitis virus is a result of recombination between Siberan and Far-Eastern subtypes" for consideration at PLOS Neglected Tropical Diseases. As with all papers reviewed by the journal, your manuscript was reviewed by members of the editorial board and by several independent reviewers. The reviewers appreciated the attention to an important topic. Based on the reviews, we are likely to accept this manuscript for publication, providing that you modify the manuscript according to the review recommendations. 

Sincerely,

Wen-Ping Guo

Academic Editor

Samuel Scarpino

Section Editor

Reviewer's Responses to Questions

**Key Review Criteria Required for Acceptance?**

**Methods**

-Are the objectives of the study clearly articulated with a clear testable hypothesis stated?

-Is the study design appropriate to address the stated objectives?

-Is the population clearly described and appropriate for the hypothesis being tested?

-Is the sample size sufficient to ensure adequate power to address the hypothesis being tested?

-Were correct statistical analysis used to support conclusions?

-Are there concerns about ethical or regulatory requirements being met?

Reviewer #1: In this revised manuscript, Sukhorukov et al. comprehensively address all provided reviewer comments in addition to suggestions for further supporting evidence/analytics. In particular, expansion of the methods section and figure/table caption text will help orient and guide readers through the logic of the presented data analyses.

Reviewer #3: (No Response)

**Results**

-Does the analysis presented match the analysis plan?

-Are the results clearly and completely presented?

-Are the figures (Tables, Images) of sufficient quality for clarity?

Reviewer #1: Minor Revisions:

Pg. 10 Par. 3 Title: (grammatical) Alternatively, consider “The Baikal subtype is evidence of recombination.”

Pg. 13 Par. 4 Line 3: Replace the reference to “node F” with “node J.” There is no node F in the revised JCF3 phylogeny (see additional Figure S2 table revision below).

Reviewer #3: (No Response)

**Conclusions**

-Are the conclusions supported by the data presented?

-Are the limitations of analysis clearly described?

-Do the authors discuss how these data can be helpful to advance our understanding of the topic under study?

-Is public health relevance addressed?

Reviewer #1: Minor Revisions:

Pg. 15 Par. 2 Line 1: How do the mentioned Figure 3 plots indicate putative recombination segment length? Do the authors mean to reference Figure 2D instead where JCF1, JCF2, and JCF3 are annotated relative to genetic position?

Pg. 16 Par 2 Line 2: No LCA value of 678 is provided in the Figure S2 table. Is the value 676 (rounded from 675.5) for the respective node B median height intended?

Reviewer #3: (No Response)

**Editorial and Data Presentation Modifications?**

Reviewer #1: Minor Revisions:

Pg. 4 Par. 3 Line 2: (grammatical) Consider “Most of these methods evaluate the strength...” instead.

Table S1: How do three asterisks differ or encapsulate the provided one and two asterisk descriptions? Alternatively, consider use of numerical superscript annotation (e.g., 1, 2, 3, etc.) instead.

Figure S2: Replace all “886-84-like” references with “Baikal subtype” to be consistent with the rest of the manuscript (if applicable).

Figure S2 (table): Remove the entry for node F in the JCF3 column. This information appears to have been moved to node J in this column based on the revised supplemental figure.

Comments & Suggestions:

Figures: Streamline the subtype-specific color-schemes between all figures. For example, the colors applied in Figure S1A do not match those applied in all the other figures.

Figure 2 Description: Provide more specific text than “in some of the alignment regions” or “for some regions” where indicated. For example, the authors may consider highlighting how indicated Joint characteristic fragments (JCFs) annotated in the GS analysis (2D) correspond to positional segments which demonstrate graphical shifts in genetic distance (2B) and/or bootstrap support (2C) to specific subtypes as compared to surrounding positional segments.

Reviewer #3: (No Response)

**Summary and General Comments**

Reviewer #1: No additional major revisions or new analyses are recommended. Provided revisions and comments are minor with a few data/graphical inconsistencies indicated within the manuscript text and figures/tables.

Reviewer #3: The authors addressed all my comments. I have no further comments on this submission.

PLOS authors have the option to publish the peer review history of their article (what does this mean?). If published, this will include your full peer review and any attached files.

Reviewer #1: No

Reviewer #3: No

Figure Files:

Data Requirements:

Reproducibility:

References

---

## [Editor Report · Decision Letter 2]

6 Feb 2023

Dear Dr. Bazykin,

We are pleased to inform you that your manuscript 'The Baikal subtype of tick-borne encephalitis virus is evident of recombination between Siberian and Far-Eastern subtypes' has been provisionally accepted for publication in PLOS Neglected Tropical Diseases.

Best regards,

Wen-Ping Guo

Academic Editor

Samuel Scarpino

Section Editor

---

## [Editor Report · Acceptance letter]

21 Mar 2023

Dear Dr. Bazykin,

We are delighted to inform you that your manuscript, "The Baikal subtype of tick-borne encephalitis virus is evident of recombination between Siberian and Far-Eastern subtypes," has been formally accepted for publication in PLOS Neglected Tropical Diseases.

Best regards,

Shaden Kamhawi

co-Editor-in-Chief

Paul Brindley

co-Editor-in-Chief
